# The Influence of Periodontal Disease on Oral Health Quality of Life in Patients with Cardiovascular Disease: A Cross-Sectional Observational Single-Center Study

**DOI:** 10.3390/medicina58050584

**Published:** 2022-04-24

**Authors:** Pompilia Camelia Lazureanu, Florina Georgeta Popescu, Laura Stef, Mircea Focsa, Monica Adriana Vaida, Romeo Mihaila

**Affiliations:** 1Department of Physiology, Faculty of Medicine, “Lucian Blaga” University, 10, Victoriei Boulevard, 550024 Sibiu, Romania; pompiliagherman@yahoo.com; 2Department of Occupational Health, “Victor Babeş” University of Medicine and Pharmacy, 2 Eftimie Murgu Square, 300041 Timisoara, Romania; 3Department of Oral Health, Faculty of Medicine, “Lucian Blaga” University, 10, Victoriei Boulevard, 550024 Sibiu, Romania; laura.stef@ulbsibiu.ro; 4Department of Medical Informatics and Biostatistics, “Victor Babeş” University of Medicine and Pharmacy, 2 Eftimie Murgu Square, 300041 Timisoara, Romania; mfocsa@gmail.com; 5Department of Anatomy and Embryology, “Victor Babeş” University of Medicine and Pharmacy, 2 Eftimie Murgu Square, 300041 Timisoara, Romania; vaida.monica@umft.ro; 6Department of Internal Medicine, Faculty of Medicine, “Lucian Blaga” University, 10, Victoriei Boulevard, 550024 Sibiu, Romania; romeo.mihaila@ulbsibiu.ro

**Keywords:** OHIP-14, cardiovascular disease, periodontal disease, quality of life

## Abstract

*Background and Objectives*: Cardiovascular disease is a leading cause of global death with a rising prevalence and a heavy economic burden. Periodontal disease has been associated with cardiovascular diseases—including incident coronary heart disease, peripheral artery disease and ischemic stroke. The study evaluates the quality of life of patients with cardiovascular and periodontal disease from the point of view of oral health by using the short version of the Oral Health Impact Profile (OHIP-14) questionnaire. *Materials and Methods*: This study included a total of 221 patients (61.86 ± 15.03 years old) selected from the Emergency Hospital of Sibiu, Romania. The participants self-completed the OHIP-14 questionnaire and they benefited from an oral health examination conducted to assess the presence and the severity of periodontal disease. *Results*: Out of the 147 patients with cardiovascular disease, 77.5% had periodontal disease (32.6% stage I, 29.2% stage II, and 15.6% stage III and IV). The presence of periodontal disease was associated with a lower oral-health-related quality of life (*p* < 0.001, ANOVA) and with a higher OHIP-14 score in patients with cardiovascular disease (18.67 ± 8.17, *p* < 0.001 ANOVA). No significant difference was observed concerning patient sex and background; however, age, body mass index and the lack of an appropriate oral hygiene routine had a strong association with the individual quality of life. The general OHIP-14 score was higher in patients with periodontal disease and associated cardiovascular disease, the presence of both cardiovascular and periodontal disease being associated with a lower quality of life. *Conclusions*: By increasing the patients’ awareness to oral healthcare measures, better outcomes and improved oral-health-related quality of life could be observed.

## 1. Introduction

Periodontal disease is an inflammatory condition affecting the gums and the supporting tissue of the teeth, leading to alveolar bone resorption and early tooth loss [1].

Although an exact cause of the development and progression of periodontal disease is not yet known, researchers agree that it is a complex disease, being a result of multiple contributing factors. An important role in the pathogeny is held by a disruption in the oral microbiome, a complex local ecosystem interfering with the dental structures [2], which can determine a local immune and inflammatory response. Sustained gingival inflammation is critical to periodontal disease progression [3]. It is important to understand the inflammatory mechanism in periodontal disease and how this inflammation may contribute both to the development of dysbiosis and, in a deregulated state, the destructive disease process [4].

Furthermore, animal model studies [5] show that periodontal bacteria and local inflammatory markers can disseminate and cause a systemic response and therefore, by causing a low-grade systemic inflammation, might influence the development of comorbidities [6]. There is evidence that systemic conditions, such as neoplastic diseases, may affect the periodontal apparatus independent of dental plaque biofilm-induced periodontitis; therefore, thorough oral hygiene and a periodical oral-health examination are required [7].

Its prevalence is reported to be between 20% and 50% around the world [8], whereas severe periodontal disease is the 11th most prevalent condition according to an analysis by the Global Burden of Disease Study 2016 [9].

Severe periodontal disease is estimated to affect 10% to 15% of the adult population worldwide [10,11]. Its severity depends both on environmental and host risk factors, and the most frequent modifiable causes of periodontal disease being reported are poor oral hygiene and tobacco use [12].

Periodontal disease is assessed by examining radiographic bone loss or by determining the clinical attachment loss by probing. The consensus recommendations [13] state that an individual case of periodontitis should be further characterized by describing the stage and grade of the disease, both permitting establishing the complexity of case management and analyzing the rate of progression, therefore choosing the best treatment protocol for the patient [14].

The management of periodontal disease includes preventative (rigorous oral hygiene, behavioral changes and risk-factor control), non-invasive treatment and supportive periodontal therapy. Oral hygiene remains an important keystone in the management of patients, and could also attenuate the cardiovascular risk [15]. Consensus was achieved on recommendations covering different interventions, such as behavioral changes, managing gingival inflammation and risk-factor control; supra- and sub-gingival interventions with and without adjunctive therapies; different types of periodontal surgical interventions; and the necessary supportive periodontal care to achieve long-term disease control [13].

The oral healthcare giver has to take into consideration any other coexisting pathologies-other health issues such as xerostomia, presence of dental plaque, caries, prosthetics and also systemic diseases that could lead to altered wound healing and altered bone physiology and could influence the treatment and the prognosis [16].

Considering the pathogenesis of periodontitis as the local inflammatory and immune response with potential systemic influence, researchers conclude the need to evaluate modulatory agents, such as monoclonal antibodies and TND alpha inhibitors with promising results in several studies [17].

Conventional treatment such as scaling and root planing, associated or not to systemic antimicrobials, have proven efficient in the prognosis of stages I–III of periodontal disease [18]. Of course, in severe cases of periodontal disease, there is need of recourse to surgical therapy, such as gingivectomy, flap debridement and excisional attachment procedures. Progress in oral health permitted the introduction of less invasive techniques, such as laser-assisted attachment procedures, with promising results [19].

Taking into consideration patients with cardiovascular disease, clinicians recommend thorough oral-health hygiene associated to mechanical debridement associated to antibiotic prophylaxis in selected cases. In some cases, a decrease of TNF alpha and cytokines was noted, therefore suggesting that the oral healthcare provider has a special place in managing patients with cardiovascular disease [20].

Patients with comorbidities are at higher risk of complications if an oral surgical procedure is performed. There are guidelines discussing the contraindications of implant rehabilitation in some cases, such as recent myocardial infarction and cerebrovascular accident, valvular prosthesis surgery, immunosuppression, bleeding issues, active treatment of malignancy, drug abuse and psychiatric illness, as well as intravenous bisphosphonate use [21].

There are numerous invasive therapeutical options for patients with periodontal disease and if an invasive treatment is considered necessary, precautions should be taken in elderly patients with multiple treatments and comorbidities [13].

Adjuvant therapies that could stimulate periodontal regeneration by using leukocytes and platelet-rich plasma associated to bone grafts prove to be efficient and seem non-inferior to sole bone grafts and collagen membrane use. By stimulating a faster and more efficient periodontal regeneration, there is hope of fewer invasive interventions, notably in elderly patients with multiple associated diseases, and therefore of a lesser risk of complications, but their long-term efficiency is yet to be proved [22].

In terms of prognosis, an untreated and/or inadequately treated periodontal disease leads invariably to the loss of tooth-supporting tissues and teeth. This issue has a major impact on patients’ quality of life from an aesthetic point of view, but also affects the chewing function, leading to further nutritional problems, especially in elderly patients, with an influence on general health [23].

Researchers point out a possible relationship between this oral-health problem and systemic diseases such as diabetes, cardiovascular disease, chronic kidney disease, malignancy and even pregnancy complications [24,25,26,27,28]. There are studies suggesting that periodontal disease may contribute to the development of cardiovascular events [29] because of the systemic inflammation generated [30], with an increase in the risk of coronary heart disease and stroke, even in younger subjects [31].

Periodontal disease also represents a current oral-health problem in Romania, with a prevalence higher than expected in both young and elderly populations, reported to be between 41% and 65% [32,33].

Patients with periodontal disease tend to have a poor oral-health-related quality of life [34,35] and by assessing patients with other associated chronic diseases, studies show that oral health might not be a major concern for these patients [36,37].

Many researchers show strong association between periodontal disease and systemic conditions and notice a positive impact of treating oral-health problems on the evolution of chronic disease [38]. The oral healthcare providers must also put emphasis on patients’ quality of life to support addressability to dental facilities and perform early treatment according to the oral conditions.

Even if dental treatment is safe when properly performed in patients with associated cardiovascular diseases [39,40], patients might have a low addressability to an oral healthcare provider [41], this leading to the aggravation of the oral-health problems and, therefore, to a lower OHRQOL (oral-health-related quality of life).

The Oral Health Impact Profile (OHIP) questionnaire was elaborated by Slade and Spencer in 1994, containing 49 items [42] based on a theoretical model developed by the World Health Organization (WHO) and adapted for oral health by Locker [43]. The goal of this questionnaire was to measure the self-reported discomfort, dysfunction and disability caused by oral diseases. In 1997, Slade developed OHIP-14, a shorter version that became widespread, translated and validated for use in many countries because of its good reliability and precision [44], including in Romania [45]. The OHIP-14 is structured in seven subdomains, which are: functional limitation, physical discomfort, psychological discomfort, physical disability, psychological disability, social disability and handicap.

The objective of our study was to evaluate factors influencing the quality of life in patients with periodontal disease and/or associated cardiovascular disease by applying the short version of the Oral Health Impact Profile 14 questionnaire (OHIP-14).

## 2. Materials and Methods

The study was conducted between June 2018 and December 2019 on patients from the Emergency Hospital in Sibiu, Romania. The study protocol was reviewed and approved by the Ethics Committee of the Hospital (10936/25 May 2018) and all the participants in the study gave their written, informed consent.

### 2.1. Study Group

We included patients from the Cardiovascular Unit and the Oral Health Department of the Emergency Hospital in Sibiu to evaluate the oral-health-related quality of life in individuals with periodontal disease and cardiovascular disease. Patient inclusion in the study (June to December 2018) took place as follows:-A total of 200 consecutive patients from the Cardiology Department with confirmed cardiovascular disease;-A total of 200 consecutive patients with no cardiovascular disease and no other known chronic diseases presenting themselves for oral evaluation at the Oral Health Department of the Emergency Hospital in Sibiu.

All patients gave their written and informed consent to participate in the study. The study steps were clearly explained to the participants:-To complete the OHIP-14 questionnaire and return it to the study conductor;-To complete a secondary questionnaire to obtain additional information about demographical data, oral health habits and other health problems;-To be revaluated by an oral healthcare professional at a later date.

A total of 400 OHIP-14 questionnaires and 400 secondary questionnaires, established by the research group, were distributed. Afterward, a sub-sequential oral-health evaluation to establish the presence and the severity of periodontal disease was programmed beginning in January 2019. Participants were called in by the healthcare professional.

Inclusion criteria:-Patients with and without cardiovascular disease; cardiovascular disease is defined as the presence of arterial hypertension, ischemic heart disease, arrythmia (permanent atrial fibrillation and flutter), valvular disease, cardiomyopathy and peripheral artery disease;-Patients being able to and willing to give their written and informed consent.

Exclusion criteria:-Patients who returned incomplete questionnaires and who did not present themselves to the oral-health evaluation;-Patients with other chronic diseases (diabetes, kidney disease, pulmonary disease and inflammatory bowel disease), malignancy, cognitive and psychiatric disorders, which could influence the OHIP response, and patients with congenital heart disease.

The algorithm of patient selection considering the inclusion and exclusion criteria is shown in Figure 1. A total of 221 adult patients (aged between 25 and 92; 61.86 ± 15.03 years old; 51.14% females, 48.86% males) were finally included in the study, out of which 66.5% (147) had cardiovascular disease.

### 2.2. Quality of Life Assessment

For the assessment of the oral-health-related quality of life, the validated Romanian version of OHIP-14 was applied. The internal consistency of the questionnaire applied to all 221 patients is high (Cronbach’s alpha = 0.88). Each OHIP-14 item reveals information about how frequently subjects experienced a specific impact in the last six months. Subjects completed the OHIP-14 with its standard ordinal format (‘never’, ‘hardly ever’, ‘occasionally’, ‘often’, ‘very often’) as a self-administered questionnaire before any dental treatment. The intensity of impact was coded according to the Likert scale, as follows: score 0 ‘never’, score 1 ‘hardly ever’, score 2 ‘occasionally’, score 3 ‘often’ and score 4 ‘very often’.

The overall OHIP-14 score and the sub-domain scores (functional limitation, physical pain, psychological discomfort, psychical disability, psychological disability, social disability and handicap) and their relationship with other variables were statistically evaluated.

### 2.3. Oral Examination

All patients included in the study were examined by one trained dentist from the Department of Oral Health, Faculty of Medicine, ‘Lucian Blaga’ University of Sibiu, to minimize evaluation biases. The dentist used a plane examination mirror (Carl Martin, Germany) and a dental probe (periodontal probe Fima instruments, 1 mm) to evaluate the presence and the severity of periodontal disease and record other oral-health problems. Intra-examiner reliability was ensured by using accurate instruments, by respecting the oral examinations methods recommended and by thoroughly recording the results for each patient.

The presence of dental plaque (debris index and calculus index) was noted, allowing the calculation of the Oral Hygiene Index-Simplified (Greene - Vermillion index) [46]. OHI-S was calculated by using six index teeth (1.1; 1.6; 2.6; 3.1; 3.6; 4.6). In the case of missing teeth, the adjacent teeth and, in the case of missing the adjacent teeth, the opposite teeth were evaluated. Oral hygiene could have an influence on periodontal disease development and could also influence the oral-health-related quality of life.

The periodontal status was evaluated according to the 2018 periodontal disease assessment recommendations [47]. Third molars, root remnants and dental implants were excluded in all measurements. The bleeding on probing (BOP) and the pocket depth (PD) were measured in all teeth. Pocket depth was measured from the gingival margin to the base of the gingival sulcus. BOP was noted if present during or immediately after the introduction of the probe in the gingival sulcus and while performing a smooth lateral move along the pocket wall on six sites per tooth (buccal and oral). Clinical attachment loss (CAL) was determined in six index teeth with the periodontal probe (first molars or second molars as substitutes, the upper right central incisor and the lower left central incisor). Measurements were recorded on four sites per tooth, from the cement-enamel junction to the base of the gingival sulcus. No radiological assessment was performed because of technical difficulties.

### 2.4. Population Characteristics

Additional information concerning the patients included in the study was obtained by applying a secondary questionnaire that was applied by the investigator and which recorded the following:-Demographic data (age, sex, social background - urban or rural);-Oral healthcare habits (visits to the dentist, scaling, brushing, use of mouthwash and dental floss);-Alcohol consumption (occasional drinking or regular alcohol consumption) and smoking habits (non-smoker or active smoker with tobacco exposure noted as pack years).

Height and weight were also recorded and the body mass index (BMI) was calculated. The patients’ diagnoses were obtained from their medical charts.

All these factors could influence the patients’ perceptions of the oral-health quality of life.

### 2.5. Data Analysis

The collected data and the OHIP-14 questionnaire applied are in concordance with the limits of previously published studies.

Univariate descriptive statistics (mean, standard deviation and standard error) was used to evaluate the sample’s characteristics.

We performed a cross-sectional observational single-center study on a total of 221 patients included. The data were analyzed by using Jasp and SPSS statistical software. The ANOVA test was used for determining the statistical significance of the differences between the means for two or more groups of patients, completed with eta squared for assessing the measure of association. The Pearson correlation analysis was also performed to determine the association between the OHIP-14 general score and other quantitative variables such as age, BMI and the number of dental elements. For nominal variables, the chi-squared test was applied, completed by the Ordinal Somers’ D test when both variables were ordinal. A threshold value for alpha of 0.05 was used for all the statistical tests.

To analyze the relationship between the quality of life (evaluated through the OHIP-14 questionnaire) and factors such as periodontal disease, cardiovascular disease, oral healthcare behavior, demographic variables or other characteristics (e.g., weight), we performed a multiple linear regression. The quality of life was included in the regression analysis as a dependent variable and as independent variables we included the following predictors: demographic data (social environment, sex, age, BMI), the presence of periodontal disease and/or cardiovascular disease, oral healthcare habits (frequency of visits to the dentist, dental scaling, dental brushing, mouthwash use, dental flossing) and other habits (smoking, coffee and alcohol consumption, physical activity level).

To predict the quality-of-life score, a regression analysis with backward elimination was conducted to determine the variables associated with the oral-health quality-of-life score. Multiple analysis models were generated (a higher R2 adjusted indicated the relevance of the analysis model). The analysis of the ANOVA variance was used to determine the influence of the independent variables on the dependent variable (quality of life).

## 3. Results

### 3.1. Study Group Characteristics and OHIP-14 General Score

The study group included 221 individuals, aged between 25 and 92 (61.86 ± 15.03), mostly living in an urban area (67%).

Out of all the patients included, 147 had cardiovascular disease and 74 were free of cardiovascular disease, as seen in Figure 2. Oral-health examination revealed that 131 (59.3%) subjects had periodontal disease (27.6% stage I periodontitis, 21.3% stage II and 10.4% stage III and IV).

Table 1 shows the population characteristics (social-demographic data, oral health problems identified, oral health care habits, the distribution of periodontal disease and physical activity level) and the association these factors might have with the general OHIP-14 score in this group of patients.

Summary scores of ordinal responses ranged from 2 to 36 points and the average overall OHIP-14 score is 12.5 ± 9.1.

In the univariate analysis, age, BMI, oral healthcare habits and the number of missing teeth were factors with significant influence on the quality of life of our group of patients as shown in Table 1. Moreover, there were significant differences between age groups; patients over 70 years old have an above-average OHIP-14 score of 17.17 (*p* < 0.001).

No significant difference in the OHIP-14 general score was found between patients residing in urban or rural areas (*p* = 0.339), nor between men and women (*p* = 0.63). Alcohol and tobacco consumption also did not seem to have an influence on the oral-health-related quality-of-life score.

The ANCOVA variance analysis was used to determine whether there is a difference of the oral-health-related quality-of-life perception in the function of the severity of the periodontal disease. The analyzed variables had a normal distribution. The skewness coefficient in absolute value was less than 1, as seen in Table 2 and the condition of the homogeneity of the variances was applied, the Levene test indicating F (3.217) = 1.97, *p* = 0.119.

Periodontal disease has an important impact on the oral-health quality-of-life perception as shown by the results of the variance analysis [F (3.216) = 1071.9, *p* = 0.000]. The intensity of the periodontal disease diagnostic on the quality of life is very increased (eta square partial _n2p is 0.937).

There is also a significant difference concerning the oral-health quality of life among patients at the different stages of periodontal disease, with an increased intensity effect (d > 1), as seen in Table 3. Patients with a severe form of periodontal disease have a lower quality of life than patients presenting an early stage of the disease.

### 3.2. OHIP-14 Score in Patients with Cardiovascular Disease

The patients with cardiovascular disease had a lower quality of life than patients with no cardiovascular condition (*p* < 0.001 with a measure of association of 22%, ANOVA). The presence of both cardiovascular disease and periodontal disease in an individual are associated with a lower quality of life (*p* < 0.001 ANOVA); results shown in Figure 3. 

The ANOVA test results revealed that the individuals of these four groups obtained significantly different results [F (3.216) = 1071.9, *p* < 0.001] for the diagnosis of periodontal disease. The magnitude of the effect of the periodontal disease diagnosis on quality of life was very large (partial square eta_η2p is 0.937). The OHIP-14 general score had a much lower value in subjects without periodontal disease (4.63 ± 1.72) compared to subjects with severe periodontal disease (30.70 ± 3.04). By applying the Gabriel test, we observed significant differences between the four groups of patients. The size of the effect in all situations was very large (d > 1).

Multiple analysis regression showed that periodontal disease, oral hygiene and BMI have a high influence on the patients’ oral-health-related quality of life. The condition of normality of the residual distribution was reached, as shown in Figure 4.

Table 4 shows the standardized and non-standardized regression coefficients for each variable. Three variables were identified to have a significant influence on oral-health quality of life: periodontal diseases, oral hygiene and BMI.

The indicators for each of the variables of the selected model of multiple analysis are the following: rsp = −0.654 for periodontal disease; rsp = −0.085 for oral hygiene (dental floss); and rsp = −0.173 for BMI.

The corresponding equation is the following:
Y = a + b1 × X1 + b2 × X2 + b3 × X3
where Y = dependent variable; a = constant; b1, b2, b3 = standardized beta coefficients; and X1, X2, X3 = independent variables.
(OHIP = −5.01 + 12.49 × periodontal disease − 1.71 × dental flossing + 0.41 × BMI).

### 3.3. OHIP-14 Domains

Assessing each OHIP-14 domain, patients in the study group scored higher in the physical pain, functional limitation and physical disability domains with seemingly less impact of psychological disability on the oral-health quality of life (Table 5).

The periodontal disease had an important impact on all patients’ oral-health-related quality of life in each OHIP-14 domain (Table 6). Patients experiencing both cardiovascular and periodontal disease (regardless of severity) scored higher (*p* = 0.001, ANOVA) in most domains. The psychological discomfort seemed to have a lesser impact on the patient’s quality of life in both patients with and without cardiovascular disease. In the group of patients with both cardiovascular and periodontal diseases, the highest OHIP-14 score was obtained in the physical pain domain, followed by the physical disability and the functional limitation domains. The patients with periodontal disease but no associated cardiovascular disease scored higher in the physical pain domain, followed by the functional limitation and the physical disability domains.

Patients with both cardiovascular disease and periodontal disease are older than patients with no periodontal disease associated, also having a higher OHIP-14 general score, as seen in Table 7.

## 4. Discussion

The present study shows that patients with both periodontal disease and cardiovascular disease have a higher OHIP-14 score compared to patients without cardiovascular disease. Understanding the patients’ perspectives on oral health and oral-health-related quality of life is an important research domain as shown by multiple published studies [48,49].

Previous studies have considered negative the impact of periodontal status in patients’ quality of life [50], severe cases of periodontal disease having a greater influence on the oral-health quality of life [51]. It reflects changes at the physiological, psychological and social levels, and it also gives an insight into a person’s overall health status [52,53]. An adequate treatment performed brings a significant improvement in individuals [54,55].

There are few studies published concerning the consequences of periodontal disease on the quality of life in patients with associated chronic diseases such as chronic kidney disease, diabetes or cardiovascular disease [56].

There are few studies in Romania approaching the oral-health-related quality of life on limited groups of patients with periodontal disease [57,58]. This is the first Romanian study to assess this subject in patients with associated cardiovascular disease, which is frequent in this population in comparison to other European countries [59] and a leading cause of death in Europe, with 20% of deaths caused by ischemic heart disease, according to recent statistics [60].

There are several oral-health quality-of-life instruments such as GOHAI (Geriatric Oral Health Assessment Index) [61] used for elderly patients and OIDP (oral impacts on daily performances) for adult and adolescent use [62]. The most frequently applied tests are OHIP with its shorter version OHIP-14, with an easy and quick application. The measurement of quality of life allows the study of the effects of a treatment or disease, the expectations of the patient, and how it is perceived according to its environment. Periodontal disease, as a prevalent oral-health problem, has economic, social and psychological consequences, and needs a multidisciplinary approach [63].

### 4.1. OHIP-14 General and Domain Score

In comparison with the European populations with periodontal disease, this group of patients had lower OHIP-14 scores in severe forms of periodontal disease, which may be a sign of reduced perception of existing oral-health problems [64,65]. This could be explained by cultural habits, by the lack of extensive educational programs to include patients of all ages and by the lack of addressability to the oral healthcare provider. Nevertheless, even with lower scores in OHIP-14 a significant impact of periodontal disease and cardiovascular disease on oral-health quality of life was observed.

The highest OHIP-14 mean subscale scores were obtained within the subscales of physical pain, followed by functional limitation and the lowest in psychological disability, in our group of patients. Our findings were similar to other studies published, Eltas et al.*ii* also found higher scores in the physical pain domain in Turkish patients with severe forms of periodontal disease [34], the same as a study published in Iasi on a smaller group of patients [58].

Periodontal status was significantly associated with lower oral-health-related quality of life, and patients with severe forms of periodontal disease had a higher general OHIP-14 score and in its domains, similar to other studies [35].

### 4.2. Oral Health Related Quality of Life in Patients with Cardiovascular Disease

The oral-health-related quality of life in patients with chronic diseases is a current discussion topic [65] including patients with cardiovascular disease and a relationship between periodontal and cardiovascular disease has been assessed in several studies [66,67].

A causal association between periodontal disease and cardiovascular disease seems difficult to conclude [68] because both of these diseases have common risk factors, but a consistent association between the two diseases has been suggested by multiple studies [69].

The oral-health status of the patients with cardiovascular disease included in the study seems to be related to the perception of the oral-health-related quality of life, in this situation, the OHIP-14 general score being 15.53 ± 9.32, higher in patients with both periodontal and cardiovascular disease (18.68 ± 8.17). The patients scored higher in the physical pain and functional limitation domains. Furthermore, we observed that several variables are associated with a lower oral-health quality of life in this group of patients with cardiovascular disease.

Older patients report a lower oral-health-related quality of life, a fact also shown by a study on a group of Romanian patients from another university clinic [41], but it does not particularly assess periodontal disease nor patients with associated cardiovascular disease. The prevalence of cardiovascular disease increases with age [69], the same in the case of periodontal disease [11]. The patients included in the study had similar ages as reported in multiple studies (70.3 ± 8.77 years) for patients with both cardiovascular and periodontal disease.

We found no significant difference related to sex in the general OHIP-14 score, nor to domain, although in some studies, men reported a lower quality of life than women in the functional limitation domain [70]. Other studies performed on a larger population also reported no significant association regarding the sex [71].

Obesity has been identified as a risk factor for the development and progression of the periodontal disease [72] and, at the same time, it represents a modifiable risk factor for cardiovascular diseases [73]. Studies also show that obesity has a negative impact on the OHRQOL of patients with periodontal disease [74], with significant differences between obese patients and patients with normal weight. According to the WHO Global Health Observatory Data Repository and Eurostat, the prevalence of obesity is expected to increase in Romania, children and young adults included. Healthcare providers could increase the awareness of the population by implementing programs addressed to all age categories to avoid complications related to this public health problem.

The OHIP-14 overall score seems not to be influenced by the patients’ residences, but higher scores in the OHIP-14 subdomains (physical pain, social, functional limitation, etc.) were obtained in patients residing in urban areas, suggesting that patients from urban areas pay more attention to their symptoms (pain, function) and they perceive that oral-health status may have a negative impact on social outcomes.

The majority of previous studies concluded that periodontal diseases are associated with a worse health-related quality of life, and this impact increases with disease severity [75]. This study confirms the significant association between severe forms of periodontal disease with high scores in physical pain and functional limitation. The overall OHIP-14 score is higher in patients with periodontal disease and associated cardiovascular disease, suggesting that a chronic associated disease has a strong association with the oral-health-related quality of life in individuals already experiencing periodontal disease.

The OHIP-14 questionnaire has a high internal consistency, being a reliable evaluation tool in patients with chronic conditions. By applying the questionnaire, individual needs would be brought to light and the necessary treatment procedures could be performed in time to avoid possible complications related to oral-health conditions. Moreover, modifiable risk factors could be identified early and the healthcare provider could sensitize his patients into adopting prevention methods.

## 5. Conclusions

Patients with periodontal disease and associated cardiovascular disease experience a poor oral-health-related quality of life with higher general OHIP-14 scores, mostly complaining about physical pain and functional limitation.

In univariate models, age, BMI, smoking habit, poor oral hygiene and tooth loss were associated with a lower oral-health-related quality of life. However, in a multivariable model, age, body mass index and oral hygiene (dental flossing) were significant.

The patients included in the study seem to have a lack of knowledge of health and oral-hygiene habits. OHIP-14 is a useful tool to help patients and oral healthcare providers to identify specific problems that could impact the quality of life. By increasing the awareness of oral healthcare measures, we could see better outcomes and improved oral-health-related quality of life and highlight the importance of a multidisciplinary approach in patients with chronic diseases.

The limitations of the study consist of the uneven sample distribution because of incomplete questionnaires and low addressability to appointments with oral healthcare professionals. This is a cross-sectional non-randomized study; the sample size was obtained by taking into consideration consecutive patients from the cardiology and oral-health departments, which fulfilled the established criteria; therefore, the results obtained apply solely to the studied population.

As future prospects, a multicentric-approach study including a larger population study could help increase patients’ addressability to oral healthcare givers. Oral-health hygiene could also be a subject for promoting health at the workplace to increase the workers’ knowledge and awareness and to diagnose periodontal diseases early.

## Figures and Tables

**Figure 1 medicina-58-00584-f001:**
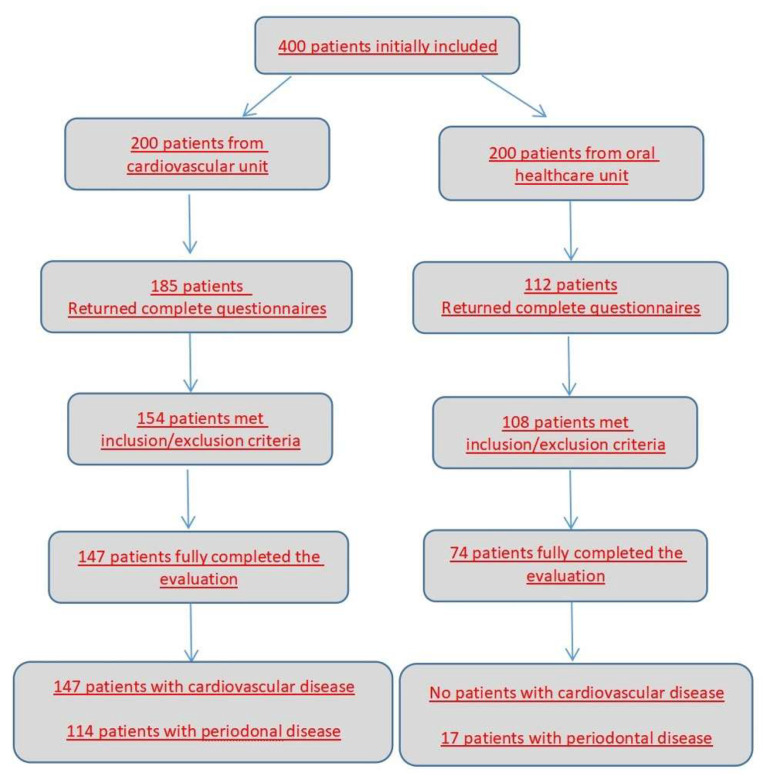
Patient selection.

**Figure 2 medicina-58-00584-f002:**
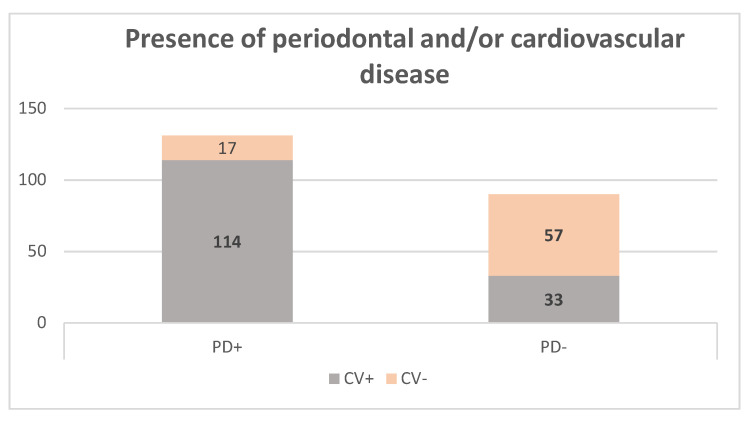
Distribution of periodontal disease and cardiovascular disease in the study group (CV+ patients with cardiovascular disease, CV− without cardiovascular disease, PD+ with periodontal disease, PD− without periodontal disease).

**Figure 3 medicina-58-00584-f003:**
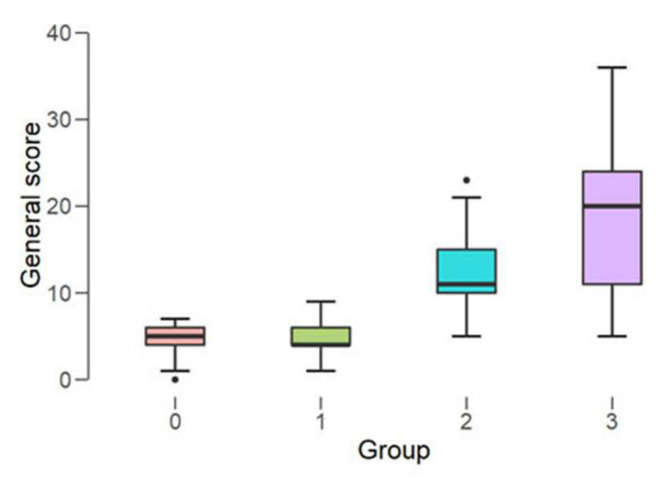
Periodontal disease and associated cardiovascular disease in association with the general OHIP-14 score (Group 0: no cardiovascular/no periodontal disease (N = 57); Group 1: cardiovascular disease only (N = 147); Group 2: periodontal disease only (N = 131); Group 3: periodontal and cardiovascular disease (N = 114)). The dots in group 0 and group 2 represent extreme values of OHIP-14 general score: 0 and 23, respectively.

**Figure 4 medicina-58-00584-f004:**
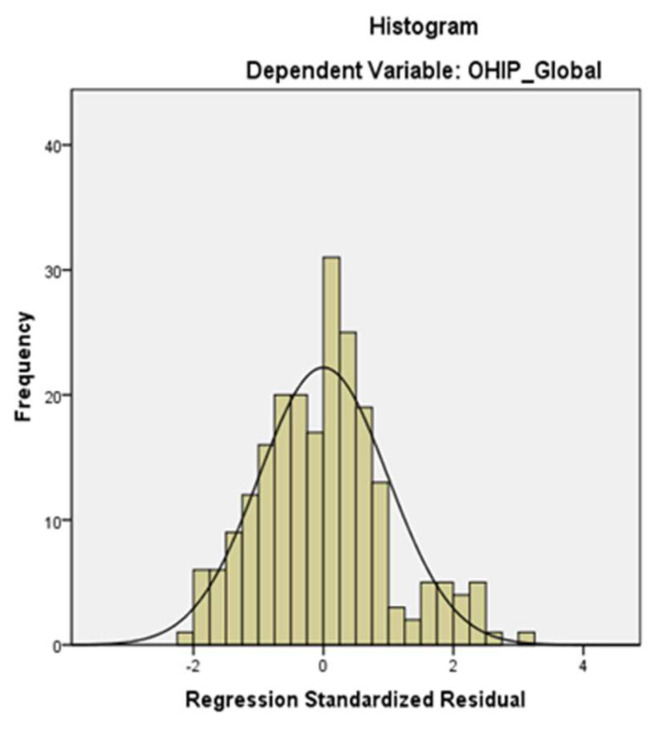
The normality of the standardized residual distribution compared to the deviations from the normal (Mean = −1.13*10^15^ Std. Dev. = 0.993 N = 221).

**Table 1 medicina-58-00584-t001:** Population characteristics and OHIP-14 general score.

Variable		Sample Distribution	OHIP-14 Score	Measure of Association	Significance *p* Value
**General Characteristics**					
Age groups *	25–50	44 (19.9%)	5.8 ± 4.2	0.188 **	
	50–70	111 (50.2%)	12.37 ± 8.03		<0.001
	70–95	66 (29.9%)	17.17 ± 10.32		
Sex	Female	113 (51.1%)	12.78 ± 9.22	0.001 **	0.63
	Male	108 (48.9%)	12.19 ± 8.99		
Social background	Rural	73 (33%)	13.33 ± 9.15	0.004 **	0.03
	Urban	148 (67%)	13.08 ± 9.07		
Education level	No education	6 (2.7%)	22.167 ± 8.998	0.138 **	
	Primary, middle and high school	124 (18.5%)	14.774 ± 9.281		0.01
	University	91 (77.8%)	8.747 ± 7.255		
BMI	<25 > = 25	87 134	9.621 ± 7.296 14.358 ± 9.679	0.336 **	0.01
Cardiovascular disease	Yes No	147 (66.51%) 74 (33.49%)	15.53 ± 9.32 8.71 ± 3.40	0.22 *	<0.001
**Oral-health status**					
Dental plaque	Yes	114 (51.6%)	13.31 ± 8.55	0.009 **	0.17
OHI-S	Good (0–0.6) Fair (0.7–1.8) Poor (1.9–3)	78 83 60	7.6 ± 7.2 11.19 ± 7.06 20.56 ± 8.43		
Missing teeth	Yes	9.2 ± 1.32	13.97 ± 9.338	0.095 **	<0.01
General periodontal disease	Absent Stage I Stage II Stage III and IV	90 (40.7%) 61 (59.3%) 47 (21.3%) 23 (10.4%)	4.59 ± 1.71	0.941 **	<0.01
**Oral healthcare habits**					
Dental evaluation	Never Rarely 1/year 2/year	30 (13.8%) 120 (55%) 61 (28%) 7 (3.2%)	16.16 ± 10.412 12.54 ± 8.9 10.94 ± 8.76 9.43 ± 4.65	0.035 *	0.05
Dental scaling	Never 1/year 2/year	163 (74.6%) 49 (22.17%) 7 (3.2%)	14.64 ± 9.61 11.19 ± 8.5 8.05 ± 6.307	0.056 *	0.002
Brushing	Rarely 1/day 2/day	3 (1.35%) 63 (28.5%) 155 (70.13%)	17.64 ± 13.31 14.73 ± 9.62 11.48 ± 8.67	0.030 **	0.03
Mouthwash	Yes	97 (43.9%)	9.64 ± 7.57	0.077 **	0.01
Flossing	Yes	73 (33%)	9.86 ± 7.83	0.041 **	0.02

* Pearson chi-square; ** eta squared.

**Table 2 medicina-58-00584-t002:** Descriptive indicators for the OHIP questionnaire in the function of the severity of the periodontal disease.

Periodontal Disease	N	Minimal OHIP-14 Score	Maximal OHIP-14 Score	Average	Standard Deviation	Symmetry Statistic—Std		Kurtosis Statistic—Std	
Absent	90	0	9	4.63	1.72	−0.239	0.254	−0.075	0.503
Stage I	61	5	15	10.67	2.45	−0.628	0.306	0.279	0.604
Stage II	47	16	26	21.40	2.145	−0.070	0.347	−0.023	0.681
Stage III and IV	23	28	36	31.18	1.9915	0.481	0.491	0.014	0.953

Legend: patient number (N); symmetry and kurtosis indicators (skewness and kurtosis with standard error Std) for the OHIP-14 questionnaire.

**Table 3 medicina-58-00584-t003:** The Gabriel test for comparing the different stages of periodontal disease.

Periodontal Disease		N	Subset			
			1	2	3	4
dimension1	Without	90	4.63			
	Stage I	61		10.67		
	Stage II	47			21.40	
	Stage III and IV	23				30.70
	Significance		1.000	1.000	1.000	1.000

Legend: N = number of patients.

**Table 4 medicina-58-00584-t004:** Beta-standardized coefficients indicating a significant influence of the independent variables on the dependent variable (quality of life).

Multiple Regression	Non-standardized Coefficients		Standardized Coefficients				Correlations	
	B	Standard Error	Beta	t	p	Zero Order	Partial	Part
Constant	−5.01	2.80		−1.79	0.08			
Periodontal diseases	12.49	0.85	0.677	14.64	0.000	0.727	0.705	0.654
Oral hygiene (dental floss)	−1.71	0.89	0.088	−1.91	0.057	−0.245	−0.129	−0.085
BMI	−0.41	0.11	0.176	3.87	0.000	0.305	0.254	0.173

**Table 5 medicina-58-00584-t005:** Average OHIP-14 domain scores in all the patients included in the study.

OHIP Domain	Mean ± SD
Functional limitation	2.15 ± 1.55
Physical pain	3.09 ± 1.81
Psychological discomfort	1.69 ± 1.56
Physical disability	2.09 ± 1.98
Psychological disability	0.82 ± 1.3
Social handicap	1.13 ± 1.28
Handicap	1.45 ± 1.23

SD = standard deviation.

**Table 6 medicina-58-00584-t006:** Average OHIP-14 domain scores in patients with and without cardiovascular disease.

OHIP-14 domain	Cardiovascular Disease		*p* Value	No Cardiovascular Disease		*p* Value
	No Periodontal Disease	Periodontal Disease		No periodontal Disease	Periodontal Disease	
Functional limitation	1 ± 0.791	3.123 ± 1.311	0.01	0.860 ± 0.811	2.353 ± 1.455	0.01
Physical pain	1.576 ± 0.969	4.167 ± 1.579	0.01	1.684 ± 0.985	3.588 ± 1.372	0.01
Psychological discomfort	2.588 ± 1.579	0.576 ± 0.792	0.01	0.579 ± 0.565	1.765 ± 1.033	0.01
Physical disability	0.576 ± 0.708	3.351 ± 1.819	0.01	0.474 ± 0.734	2.000 ± 1.541	0.01
Psychological disability	0.030 ± 0.174	1.535 ± 1.488	0.01	0.035 ± 0.186	0.235 ± 0.437	0.01
Social handicap	0.303 ± 0.585	1.789 ± 1.366	0.01	0.281 ± 0.453	1.353 ± 1.057	0.01
Handicap	0.606 ± 0.496	2.123 ± 1.298	0.01	0.632 ± 0.555	1.588 ± 0.712	0.01

**Table 7 medicina-58-00584-t007:** OHIP-14 general score and demographic variables in patients with and without periodontal disease.

Variables	Cardiovascular Disease		No Cardiovascular Disease	
	No Periodontal Disease	Periodontal Disease	No periodontal Disease	Periodontal Disease
Age	65.30 ± 9.876	70.30 ± 8.777	44.46 ± 12.740	56.94 ± 12.760
BMI	27.04 ± 4.072	27.22 ± 3.927	24.58 ± 3.125	24.70 ± 3.896
OHIP general score	4.667 ± 1.848	18.675 ± 8.173	4.544 ± 1.648	12.882 ± 5.171

## Data Availability

Not applicable.

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
