# Peer review of "The Influence of Periodontal Disease on Oral Health Quality of Life in Patients with Cardiovascular Disease: A Cross-Sectional Observational Single-Center Study"

_medicina, 2022, doi:10.3390/medicina58050584_

Round 1

Reviewer 1 Report

The present study is a very interesting one, which studies the relationship between cardiovascular pathologies and periodontal disease, a topic intensely debated in recent years. However, followings should be considered:

  1. Although the relationship between periodontal and cardiovascular diseases has been extensively studied recently, including on the oral health-related quality of life side, bibliographic research is weak. Only 18 out of 46 bibliographic sources are from the last 5 years. A revision of the bibliography with the inclusion of several recent sources is preferable to increase the quality of the paper. Suggestions: Molania T, and all. BMC Oral Health. 2021 Aug 11;21(1):391. 2. Park SY, et all. Eur Heart J. 2019 Apr 7;40(14):1138-1145. 3. Rodean IP, et al. J Clin Med. 2021 Mar 21;10(6):1290.
  2. Since you have studied several groups of patients, without comparing the results with those of the group with periodontal disease and concomitant cardiovascular pathologies, the formulation of the objective/s is wrong. Please reformulate.
  3. In the materials and methods section, you specify that patients with arrhythmias have been included. As a basic cardiovascular disease - as a first diagnosis? On what substrate the arrhythmia appeared? Has the type of arrhythmia been checked? A paroxysmal arrhythmia on a structural normal heart could be a source of bias and the patients should have been eliminated.
  4. Fewer patients included in the study in Oral healthcare unit had periodontal disease. Knowing that the risk of developing periodontal disease, but also cardiovascular disease increases with age, it would have been interesting to provide us comparative demographic data (especially age) between the two inclusion sites, and not only the age data on the entire population.
  5. The phrase in the lines 154-155 should be relocated to another section. Not applicable to the Methods section.
  6. I recommend reporting “p” values larger than 0.01 to two decimal places. Please re-check all “p” values in the text and in tables.
  7. In Table 1, please provide the mean +/- SD or SEM of the OHIP-14 score for patients without cardiovascular disease.
  8. Please provide in section 3.2 of the results the mean +/- SD or SEM of the OHIP-14 score, just the presence of the p value is not enough. The correctness of the statistical tests cannot be evaluated otherwise.
  9. The paragraph in lines 318-321 of the discussion section is not sufficiently substantiated in terms of bibliographic sources. Please reconsider replacing / adding new bibliographic sources.
  10. Given that the main objective of the study was to evaluate the oral health quality of life in patients with cardiovascular disease, please reconsider section Oral health related quality of life in patients with cardiovascular disease of the discussions. Introduce several studies to discuss the interrelationship between periodontal disease - cardiovascular disease AND/OR oral health related quality of life. The section mainly discusses the demographic data with very little implications related to cardiovascular disease.

Author Response

Thank you for taking in consideration our manuscript and for all the observations you made, helping us to increase the quality of our papers.

We hope that the changes we made meet your expectations.

Your sincerely, 

The Authors

Reviewer 2 Report

The article entitled “The influence of periodontal disease on oral health quality of  life in patients with cardiovascular disease” aimed to investigate the quality of life of patients with cardiovascular and periodontal disease from the point of view of oral-health, by using the short version of the Oral Health Impact Profile (OHIP-14) questionnaire. The paper is in line with journal’s aim, moreover, Authors have well revised several issues. However, I ask authors to add some key concepts.

  • In the title it is necessary to specify the type of study
  • The introduction section is too short, the authors must introduce the issue of periodontitis under various aspects, from the difficulty in the diagnosis, treatment and prognosis of the dental elements, in fact, non-surgical therapy is sometimes not enough and surgical therapy is used when the pocket has a depth greater than 5mm and is bleeding (please see and discuss DOI: 10.1002 / JPER.20-0305) and to perform surgery it is necessary that the systemic conditions of the candidate patient are optimal, hence the problem of the link between periodontitis and cardiovascular diseases, it is therefore necessary to better discuss the choice of the topic addressed.
  • Figures 1 and 2 are too grainy, they must be replaced with a better quality image
  • How was the sample size of the study calculated?
  • Research suggests how patients and clinicians define the concept of success differently.

Over the years, numerous questionnaires have been used to ascertain the level of patient satisfaction and the oral health-related quality of life (OHRQoL). Among these, the Oral Health Impact Profile (OHIP) is the most widely used. The original version included 49 questions on 7 different topics about function, social aspects, satisfaction, etc.  Please, discuss better this issue.

  • Conclusions cannot be reduced to a sentence: you must improve them highlighting the limits and the future insights pointed out from this article.

Author Response

Thank you for taking in consideration our manuscript and for all the observations you made, thus helping us increased the quality of our paper.

We hope that the changes we made meet your expectations.

Your sincerely, 

The Authors

Round 2

Reviewer 1 Report

From my point of view, the changes made by the authors are sufficient for the manuscript to be considered for publication in its current form.

Reviewer 2 Report

The authors adequately addressed this reviewer's comments